# Sequence variation at *ANAPC1* accounts for 24% of the variability in corneal endothelial cell density

Erna V. Ivarsdottir [1,2], Stefania Benonisdottir[1], Gudmar Thorleifsson[1], Patrick Sulem [1], Asmundur Oddsson [1], Unnur Styrkarsdottir [1], Snaedis Kristmundsdottir[1], Gudny A. Arnadottir [1], Gudmundur Thorgeirsson[1,3,4], Ingileif Jonsdottir[1,3,5], Gunnar M. Zoega[6], Unnur Thorsteinsdottir[1,3], Daniel F. Gudbjartsson [1,2], Fridbert Jonasson[3,6], Hilma Holm[1] & Kari Stefansson [1,3]

The corneal endothelium is vital for transparency and proper hydration of the cornea. Here, we conduct a genome-wide association study of corneal endothelial cell density (cells/mm$^2$), coefficient of cell size variation (CV), percentage of hexagonal cells (HEX) and central corneal thickness (CCT) in 6,125 Icelanders and find associations at 10 loci, including 7 novel. We assess the effects of these variants on various ocular biomechanics such as corneal hysteresis (CH), as well as eye diseases such as glaucoma and corneal dystrophies. Most notably, an intergenic variant close to *ANAPC1* (rs78658973[A], frequency = 28.3%) strongly associates with decreased cell density and accounts for 24% of the population variance in cell density ($\beta = -0.77$ SD, $P = 1.8 \times 10^{-314}$) and associates with increased CH ($\beta = 0.19$ SD, $P = 2.6 \times 10^{-19}$) without affecting risk of corneal diseases and glaucoma. Our findings indicate that despite correlations between cell density and eye diseases, low cell density does not increase the risk of disease.

[1] deCODE genetics/Amgen, Reykjavik, Iceland. [2] School of Engineering and Natural Sciences, University of Iceland, Reykjavik, Iceland. [3] Faculty of Medicine, School of Health Sciences, University of Iceland, Reykjavik, Iceland. [4] Division of Cardiology, Department of Internal Medicine, Landspitali, The National University Hospital of Iceland, Reykjavik, Iceland. [5] Department of Immunology, Landspitali, The National University Hospital of Iceland, Reykjavik, Iceland. [6] Department of Ophthalmology, Landspitali, The National University Hospital of Iceland, Reykjavik, Iceland. Correspondence and requests for materials should be addressed to H.H. (email: hilma.holm@decode.is) or to K.S. (email: kari.stefansson@decode.is)

Corneal diseases are among the most common causes of visual loss worldwide and endothelial cell failure is the leading indication for corneal transplantation[1]. The corneal endothelium is the monolayer of cells at the innermost surface of the cornea. Through an ion pump function, the endothelium is responsible for balanced corneal hydration, thus maintaining transparency by preventing edema and disruption of lamellar spacing of the collagen fibrils in the corneal stroma[2]. Maintaining proper function of the endothelium requires a minimum number of endothelial cells, around 400–500 cells/mm² [3]. The cells are generally thought to be incapable of mitosis after birth but are halted in the G1 phase of the cell cycle and their number decreases with age[2]. The cell division is thought to be blocked by contact inhibition, high concentration of negative growth factors in the anterior chamber and by accumulation of reactive oxygen species promoting state of stress-induced senescence[3,4]. The response to cell loss includes spreading and/or migration of adjacent cells which increase in size and become more variable in both cell size and shape[3,5].

Non-contact auto-tracking and focusing specular microscopy provides a non-invasive analysis of the structure of the corneal endothelium[6]. The equipment captures an image of the endothelial cell layer and provides measures of its structure including, cell density (cells/mm²), coefficient of cell size variation (CV), percentage of hexagonal shaped cells (HEX) and central corneal thickness (CCT). CCT has been studied extensively in large genome-wide association studies (GWAS) where the measurements are obtained using different types of instruments[7–13]. Several CCT associating loci have been identified including WNT10A, COL5A1, and ZNF469[11]. To our knowledge, however, there are no reports of sequence variants influencing other direct measures of endothelial structure such as cell density, CV and HEX.

The measures of endothelial structure are used to diagnose corneal diseases. Several corneal diseases including Fuchs endothelial corneal dystrophy (FECD) and macular corneal dystrophy (MCD) are known to have genetic components[14,15]. FECD is a leading cause of corneal transplant surgery and is characterized by premature loss of endothelial cells resulting in increased variability in cell shape and size leading to corneal edema and visual loss[16,17]. MCD is characterized by progressive spotted corneal opacities leading to severe visual impairment, caused by homozygous or compound heterozygous variants in the CHST6 gene (OMIM: 605294). MCD is a rare condition worldwide, but because of the founder effect[18], MCD is unusually common in Iceland where it accounts for approximately one-third of corneal transplantations[19].

Cell density in the corneal endothelium may be reduced in glaucoma patients[20]. Glaucoma is an ocular disease that affects ~3.5% of people over 40 years of age, and is a major cause of irreversible blindness worldwide[21,22]. Commonly, elevated intraocular pressure (IOP) leads to progressive damage of the optic nerve causing visual loss and it has been postulated that elevated IOP affects the endothelium[23]. The relationship between different corneal measures and glaucoma have been investigated, especially the role of IOP, CCT and more recently corneal hysteresis (CH) and corneal resistance factor (CRF)[24]. CH and CRF are measures of corneal response to a rapid jet of air, where CH is a measure of the elasticity of the cornea and CRF is an overall indicator of the resistance of the cornea[25]. Lower CH has been associated with faster rate of glaucoma progression[26–28].

Here we describe our search for sequence variants associating with measures of corneal structure, finding 10 sequence variants, nine common and one low-frequency (minor allele frequency (MAF) <5%), associating with CCT, cell density, CV, or HEX. Two of these variants satisfy thresholds for genome-wide significance for more than one trait. Seven of the associations are novel, two are represented by coding variants and one is located in a gene known to cause a Mendelian disorder (ADAMTS17, OMIM: 607511). We assess the effects on ocular biomechanics, including IOP and CH, and various eye diseases. We find a variant near ANAPC1 that strongly associates with cell density but not with risk of eye disease, which indicates that low cell density alone does not affect disease development directly.

## Results

**Summary of the data.** Endothelial images from a specular microscopy of 6125 Icelanders were used in the analysis, providing measures of cell density, CV, and HEX. The equipment also produced CCT measurements. These images were obtained as a part of a comprehensive phenotyping of a general population sample (the deCODE health study), currently including 6300 Icelanders that were between 18 and 94 years of age at the time of recruitment (44.6% men; mean age = 55.9, standard deviation (SD) = 15.1). We also measured several ocular biomechanics such as CH, CRF, Goldmann correlated intraocular pressure (IOPg), and corneal compensated intraocular pressure (IOPcc) (Supplementary Figures 1–6).

The number of endothelial cells declines with age and remaining cells enlarge to compensate for the cell loss. Consequently, cell density and HEX decrease with age while CV increases (Table 1, Fig. 1a–c and Supplementary Figure 7). Women have considerably lower HEX than men (50.6% vs 48.3%, $P = 8.4 \times 10^{-43}$; F test), while cell density and CV are slightly higher for women (2663 vs 2639 cells/mm², $P = 1.8 \times 10^{-3}$ (F test) and 30.3 vs 29.6, $P = 2.6 \times 10^{-6}$ (F test), respectively). To our knowledge, the gender differences of HEX, cell density and

### Table 1 Summary of data

| | Mean (SD) | Men's mean (SD) | Women's mean (SD) | Sex effect | Sex P-value | Glaucoma effect [SD] | Glaucoma P-value |
|---|---|---|---|---|---|---|---|
| CD | 2652 (296) | 2639 (296) | 2663 (294) | 23.83 | $1.8 \times 10^{-3}$ | −0.35 | $2.1 \times 10^{-6}$ |
| CV | 30.0 (5.9) | 29.6 (6.7) | 30.3 (5.1) | 0.71 | $2.6 \times 10^{-6}$ | 0.13 | 0.069 |
| CCT | 563 (40) | 565 (40) | 562 (39.5) | −3.82 | $1.9 \times 10^{-4}$ | −0.25 | $4.7 \times 10^{-4}$ |
| HEX | 49.3 (6.5) | 50.6 (6.6) | 48.3 (6.3) | −2.29 | $8.4 \times 10^{-43}$ | −0.16 | 0.030 |
| CH | 10.4 (1.2) | 10.2 (1.2) | 10.5 (1.1) | 0.28 | $5.7 \times 10^{-22}$ | −0.37 | $3.8 \times 10^{-7}$ |
| IOPg | 14.7 (3.4) | 14.6 (3.5) | 14.7 (3.4) | 0.13 | 0.15 | 0.18 | 0.014 |
| IOPcc | 15.3 (3.1) | 15.4 (3.2) | 15.2 (3.0) | −0.19 | 0.015 | 0.29 | $7.3 \times 10^{-5}$ |
| CRF | 13.1 (1.5) | 13.0 (1.5) | 13.3 (1.5) | 0.29 | $5.7 \times 10^{-14}$ | −0.07 | 0.34 |

The mean and standard deviation (SD) is shown for each corneal trait obtained from the specular microscopy equipment and the ocular response analyzer, overall and separately for each sex. The effect of sex and glaucoma status on each trait and the corresponding P-values (F test) are shown. The sample size was 6125 in total, 2733 men and 3392 women
CD cell density, CV coefficient of cell size variation, CCT central corneal thickness, HEX percentage of hexagonal cells, CH corneal hysteresis, IOPg Goldmann correlated intraocular pressure, IOPcc corneal compensated intraocular pressure, CRF corneal resistance factor

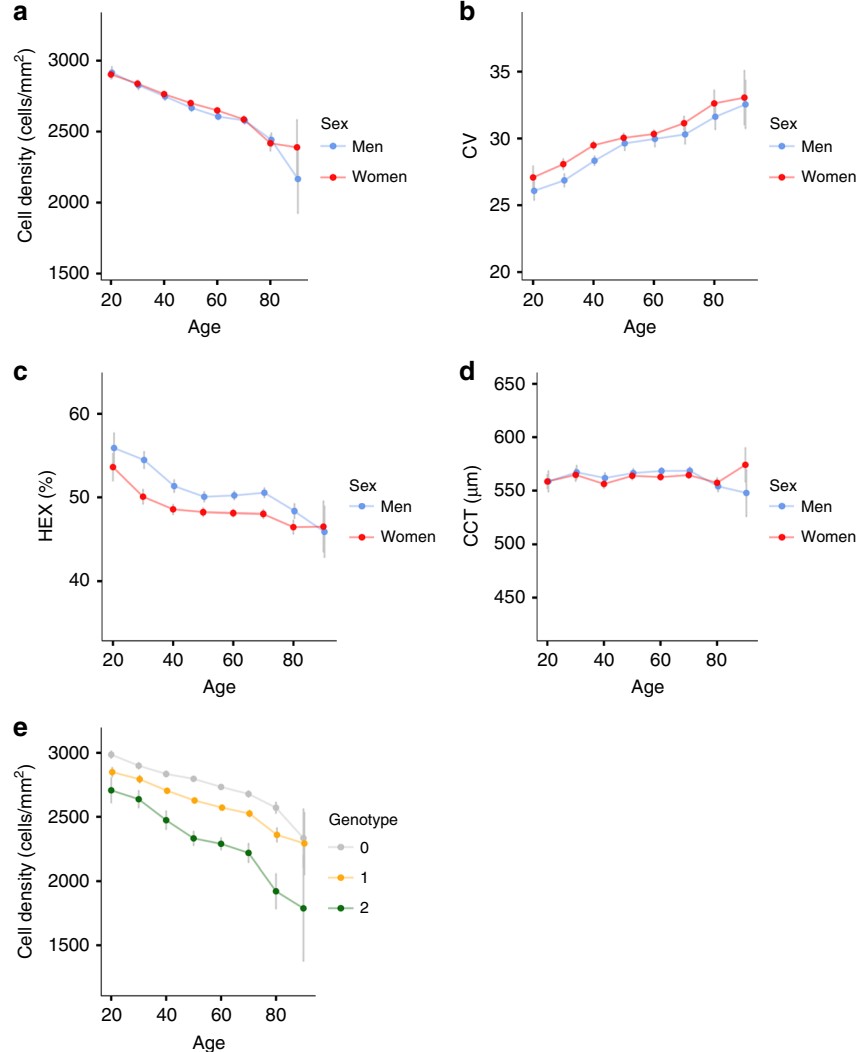

**Fig. 1** Corneal structure measurements by age ($N = 6125$). The average **a** cell density, **b** HEX, **c** CV, and **d** CCT values for subjects belonging to a 10 year age group (e.g., age = 30 for individuals between 26 and 35) against age, for men and women. **e** The average cell density for subjects belonging to a 10 year age group against age for noncarriers, heterozygous, and homozygous carriers of the *ANAPC1* variant, rs78658973. The gray lines show the 95% confidence intervals

CV have not been reported before. Consistent with previous reports[29], CCT does not change with age and is higher for men than women (565.4 vs 561.5 μm, $P = 1.9 \times 10^{-4}$; F test) (Fig. 1d). The measurements of corneal structure are correlated, also after adjusting for sex and age (Supplementary Table 1). The strongest correlations are between CV and HEX ($r = -0.65$) and between CV and cell density ($r = -0.33$).

**Study design**. To search for sequence variants associating with corneal structure, we analyzed 35.2 million sequence variants identified through whole-genome sequencing of 28,075 Icelanders that were subsequently imputed into 155,250 chip-typed individuals, as well as their first- and second-degree relatives[30,31] (Supplementary Figures 8–11). Ten sequence variants, satisfied our genome-wide significance thresholds that are dependent on sequence variant annotation[32] (Table 2, Supplementary Table 2).

To determine whether the associating variants affect the risk of eye diseases we performed a meta-analysis of GWAS results from Iceland and the UK Biobank (Supplementary Table 3 and 4). We tested the variants for association with glaucoma (8432 cases and 641,353 controls) and the sub-categories: primary open-angle glaucoma (2296 cases and 705,937 controls), and primary angle

closure glaucoma (777 cases and 637,017 controls). We also tested the variants for association with disorders of cornea (ICD10 code H18, 756 cases and 663,218 controls) and the sub-categories; corneal degeneration (ICD10 code H18.4, 199 cases and 684,021 controls), hereditary corneal dystrophies (ICD10 code H18.5, 330 cases and 683,652 controls) and keratoconus (ICD10 code H18.6, 127 cases and 659,503 controls). We applied a Bonferroni corrected $P$-value threshold based on testing ten corneal structure variants for association with seven phenotypes ($P < 0.05/(7*10) = 7.1 \times 10^{-4}$).

**GWAS results**. Two sequence variants associate with cell density (Fig. 2a, Table 2). The strongest association is represented by a common intergenic variant located 0.4 kb downstream of *ANAPC1*, rs78658973[A] (MAF = 28.3%), that associates with decreased cell density ($\beta = -0.77$ SD, $P = 1.8 \times 10^{-314}$; a likelihood-ratio test was performed in all genome-wide associations) (Fig. 1e, Supplementary Figure 12). rs78658973 is highly correlated with 113 variants ($r^2 > 0.8$) in the region, none of which is protein coding. The most highly correlated coding variants are two splice region variants in *ANAPC1*; rs201128688 and rs142711068 ($r^2 = 0.72$ and 0.73, respectively). The effect of rs78658973[A] conditioning on the two splice region

**Table 2 Association results**

| Trait | Chr:Position | rs-name | Allele (min/maj) | MAF (%) | Gene/ [Locus] | Coding effect | LD class | P-value | β [SD] (95% CI) | Ref. |
|---|---|---|---|---|---|---|---|---|---|---|
| CD | 2:111726948 | rs78658973 | (A/T) | 28.3 | [ANAPC1] | Intergenic | 112 | $1.8 \times 10^{-314}$ | −0.77 (−0.77,−0.77) | |
| CD | 18:55586154 | — | CTG repeat > 33 | 6.1 | TCF4 | | 3 | $1.4 \times 10^{-20}$ | −0.41 (−0.49,−0.32) | 35 |
| CV | 2:111726948 | rs78658973 | (A/T) | 28.3 | [ANAPC1] | Intergenic | 112 | $2.8 \times 10^{-28}$ | 0.23 (0.19,0.27) | |
| CV | 17:14650919 | rs2323458 | (A/G) | 36.1 | 17p12 | Intergenic | 47 | $6.9 \times 10^{-13}$ | 0.14 (0.10, 0.18) | |
| CV | 8:9943404 | rs10094779 | (G/A) | 24.1 | 8p23.1 | Intergenic | 4 | $7.6 \times 10^{-12}$ | 0.15 (0.11, 0.19) | |
| CV | 11:122029470 | rs76561503 | (C/T) | 17.9 | 11q24.1 | Intergenic | 29 | $3.3 \times 10^{-10}$ | 0.16 (0.11,0.21) | |
| CCT | 16:88302168 | rs12719930 | (G/A) | 39.4 | [ZNF469] | Intergenic | 21 | $1.9 \times 10^{-14}$ | −0.15 (−0.19,−0.11) | 13 |
| CCT | 2:218890289 | rs121908120 | (A/T) | 2.6 | WNT10A | Missense | 6 | $4.5 \times 10^{-11}$ | −0.39 (−0.51,−0.28) | 9 |
| CCT | 9:134545337 | rs943423 | (G/A) | 27.2 | [COL5A1] | Intergenic | 0 | $6.1 \times 10^{-11}$ | −0.14 (−0.19,−0.10) | 13 |
| CCT | 15:100152748 | rs72755233 | (A/G) | 13.8 | ADAMTS17 | Missense | 0 | $1.3 \times 10^{-10}$ | 0.18 (0.12, 0.23) | |
| CCT | 12:104015054 | rs117801489 | (C/T) | 4.3 | GLT8D2 | Missense | 2 | $3.9 \times 10^{-10}$ | 0.30 (0.20, 0.39) | |
| HEX | 18:55586154 | — | CTG repeat > 33 | 6.1 | TCF4 | | 3 | $5.9 \times 10^{-18}$ | −0.37 (−0.45,−0.28) | 35 |
| HEX | 2:111726948 | rs78658973 | (A/T) | 28.3 | [ANAPC1] | Intergenic | 112 | $2.8 \times 10^{-13}$ | −0.16 (−0.20,−0.11) | |

The 10 variants identified in the GWAS on cell density (CD), CV, HEX, and CCT. Effects are shown for the minor allele. Minor allele frequency in the Icelandic population is presented. The LD class column shows the number of highly correlated variants ($r^2 > 0.8$). The imputation information for all these variants is > 0.99
*CD* cell density, *CV* coefficient of cell size variation, *CCT* central corneal thickness, *HEX* percentage of hexagonal cells, *MAF* minor allele frequency, *LD* linkage disequilibrium, *CI* confidence interval

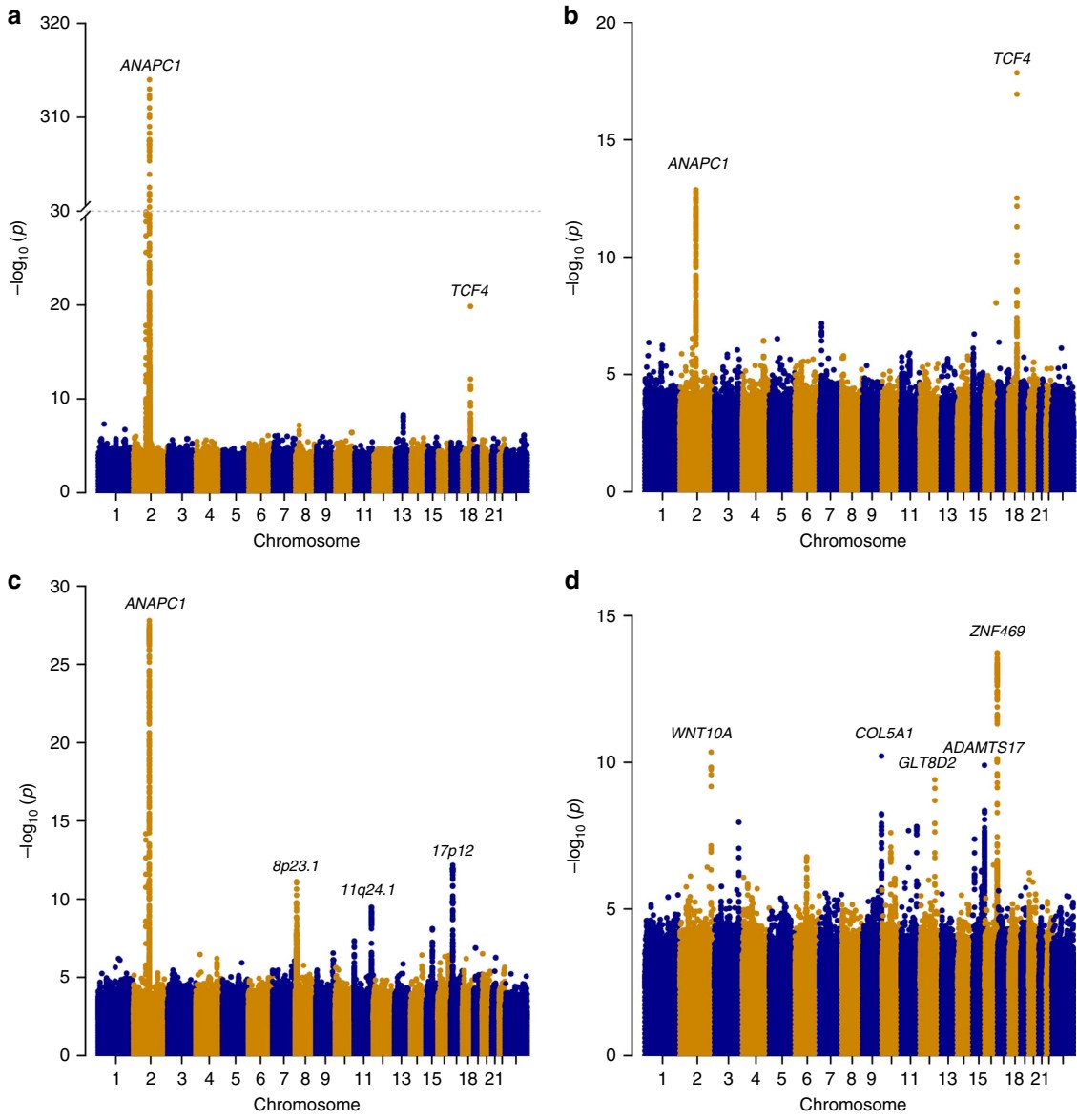

**Fig. 2** Manhattan plots. Association results for the corneal measures ($N = 6125$) obtained from the specular microscopy; **a** cell density, **b** HEX, **c** CV, and **d** CCT. The $-\log_{10}$ P-values are plotted for each variant against their chromosomal position. A likelihood-ratio test was used when testing for association

variants is still significant ($\beta = -0.78$ SD, $P = 7.8 \times 10^{-85}$), while the effect of the splice region variants are completely explained by rs78658973. rs78658973[A] also associates with CV ($\beta = 0.23$ SD, $P = 2.8 \times 10^{-28}$) and HEX ($\beta = -0.16$ SD, $P = 2.6 \times 10^{-19}$), but this is largely driven by the strong effect on cell density (CV adjusted for cell density: $\beta = -0.04$ SD, $P = 0.049$; HEX adjusted for cell density: $\beta = -0.04$ SD, $P = 0.072$). Interestingly, rs78658973[A] also associates with CH ($\beta = 0.19$ SD, $P = 2.6 \times 10^{-19}$) (Supplementary Table 4, Fig. 3) which reflects the cornea's ability to absorb and dissipate energy[25,33]. CH and cell density are only weakly correlated after adjusting for sex and age ($r = -0.03$) and the effect of rs78658973[A] on CH is not affected by adjustment for cell density ($\beta_{adjusted} = 0.21$ SD, $P_{adjusted} = 1.5 \times 10^{-18}$). CH is lower in patients with glaucoma or corneal disorders like keratoconus[34]. However, rs78658973 does not associate with corneal diseases or glaucoma in our data ($P > 0.03$, Supplementary Table 4).

The other variant associating with cell density is a common allele of a microsatellite at TCF4 (MAF = 6.1%), a CTG repeat of length $\geq 33$, corresponding to the expanded CTG 18.1 allele (OMIM: 602272, allelic variant 0.0007). This microsatellite is pathogenic according to Clinvar and has been reported to strongly predispose to autosomal dominant FECD[35–38], a disease of the corneal endothelium that affects roughly 4% of people over 40 years old (OMIM: 602272). Consistent with the characteristics of FECD, the expanded CTG 18.1 allele associates with lower cell density and HEX ($\beta = -0.38$ SD, $P = 1.6 \times 10^{-19}$ and $\beta = -0.37$ SD, $P = 5.9 \times 10^{-18}$, respectively). Interestingly, it also associates with decreased CH, CRF and IOPg ($\beta = -0.29$ SD, $P = 3.1 \times 10^{-12}$, $\beta = -0.30$ SD, $P = 7.9 \times 10^{-13}$ and $\beta = -0.18$ SD, $P = 1.7 \times 10^{-5}$, respectively) while not affecting glaucoma risk (OR = 0.92, CI = (0.82;1.03), $P = 0.15$). Consistent with previous reports, the expanded CTG 18.1 allele associates with hereditary corneal dystrophies in our data (OR = 7.7, $P = 3.3 \times 10^{-31}$, Supplementary Table 4).

The GWAS on HEX revealed only the two variants at ANAPC1 and TCF4 (Table 2, Fig. 2b), both of which associate more strongly with cell density.

We identified three loci associating most strongly with CV (Fig. 2c, Table 2). At 17p12, an intergenic variant rs2323458[A] located ~ 300 kb downstream of HS3ST3B1, associates with increased CV (AF = 36.1%, $\beta = 0.14$ SD, $P = 6.9 \times 10^{-13}$). rs2323458 is in moderate linkage disequilibrium (LD) with rs2323457 ($r^2 = 0.61$) which has been reported to associate with CCT[11]. In our data, rs2323457 also associates with CV ($\beta = 0.14$ SD, $P = 1.4 \times 10^{-10}$) but conditional analysis revealed that the effect is driven by rs2323458 ($\beta_{adjusted} = 0.043$ SD, $P_{adjusted} = 0.21$). We did not replicate the effect of rs2323457 on CCT ($\beta = -0.033$ SD, $P = 0.13$), even though the power for replication is 94% at a two-sided significance level of 0.05. Two more variants associate with CV: rs10094779[G] at 8p23.1 ~ 40 kb upstream of MIR124-1 and rs76561503[C] at 11q24.1 ~ 60 kb downstream of MIR100HG (AF = 24.2%, $\beta = 0.15$ SD, $P = 7.6 \times 10^{-12}$ and AF = 17.9%, $\beta = 0.16$ SD, $P = 3.3 \times 10^{-10}$, respectively). Variants at 8p23.1, between MIR124-1 and MSRA, have been reported to associate with high myopia[39]. rs10094779 is only moderately correlated with the reported variants ($r^2 < 0.25$). Interestingly, a variant in MIR100HG, rs577948, has also been reported to associate with myopia[40]. However, rs76561503 is not correlated with the reported variant ($r^2 = 0.03$).

Five variants associated with CCT in our data (Fig. 2d, Table 2). Three are at established CCT loci: ZNF469, WNT10A, and COL5A1[8,9,41]. The two novel CCT associations are with the missense variants p.Thr446Ile in ADAMTS17 (MAF = 13.8%, $\beta = 0.18$ SD, $P = 1.3 \times 10^{-10}$) and p.Tyr24Cys in GLT8D2 (MAF = 4.3%, $\beta = 0.30$ SD, $P = 3.9 \times 10^{-10}$). P. Thr446Ile in ADAMTS17 has been associated with decreased intraocular pressure[42] and decreased height[43]. P.Thr446Ile associates with decreased intraocular pressure in our data ($\beta = 0.14$ SD, $P = 5.8 \times 10^{-7}$), but after adjusting for CCT the association is much weaker ($\beta_{adjusted} = 0.06$ SD, $P_{adjusted} = 0.024$). Rare sequence variants in ADAMTS17 cause autosomal recessive Weill-Marchesani syndrome, a rare connective tissue disorder with features including microspherophakia, severe myopia, glaucoma, cataract, and short stature (OMIM: 607511). Notably, p.Tyr24Cys in GLT8D2 associates with increased height in a meta-analysis of the Icelandic and UK Biobank data ($\beta = 0.06$ SD, $P = 1.4 \times 10^{-11}$, N = 490,381). To understand the relationship between height and CCT, we evaluated the correlation between the effect on height in the Icelandic and UK Biobank data (N = 490,381) of 693 reported adult height variants[44] and their effects on CCT, but found no correlation ($r^2 = 0.006$; $P = 0.038$; F test) (Supplementary Figure 13). To validate the CCT association of the two novel variants, we tested them in a non-overlapping sample of 1459 Icelanders with CCT measurements from the Reykjavik Eye Study[45]. At a significance threshold of $P < 0.05$, we replicated the associations for both p.Thr446Ile in ADAMTS17 and p.Tyr24Cys in GLT8D2 with CCT ($\beta = 0.25$ SD, $P = 7.3 \times 10^{-3}$ and $\beta = 0.11$ SD, $P = 0.045$, respectively) (Supplementary Table 5).

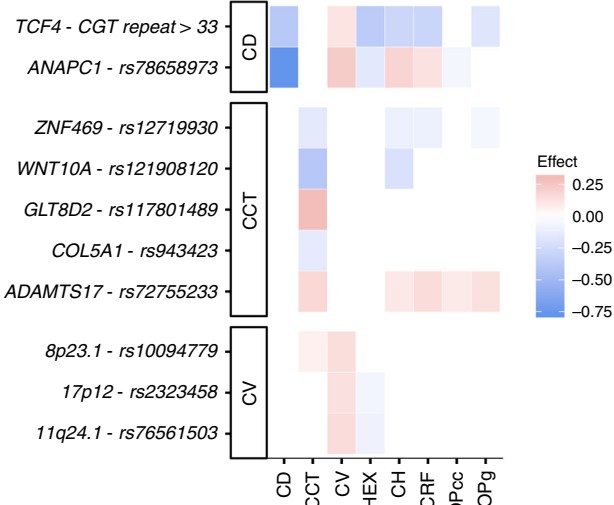

**Fig. 3** Effects of the GWS corneal structure variants on different corneal measures. Each row shows the estimated effect of the minor allele on the corneal measures from the specular microscopy and the ocular response analyzer. The variants are annotated with their corresponding gene and grouped by their strongest associating trait. The effect is shown only for significant associations after adjusting for multiple testing with a false discovery rate procedure for each variant. Red color represents a positive effect on the corneal measures and blue color represents a negative effect. Non-significant associations are colored white

**Gene expression.** We examined the expression levels of the genes at the 10 associating loci using the publicly available Ocular Tissue Database[46]. For non-coding variants we looked for all genes in a 500 kb region around the associating variant. We found that 13 of the 15 genes at the 10 loci are expressed in all ocular tissues (Supplementary Data 1). MIR124-1 and MIR100HG were not found in the database, but previous studies have reported that MIR124-1 is expressed in the human lens[47] and both MIR124-1

and *MIR100HG* are expressed in the human retina[40,48]. Expression levels varied across tissues for some genes, e.g., *GLT8D2* is most highly expressed in the cornea, *TCF4* in the sclera and *ANAPC1* in the optic nerve head.

**Disease variants affect corneal structure**. Two different mutations in *CHST6* are known to cause MCD in Iceland; a missense variant p.Ala128Val (MAF = 0.66%) and a frameshift variant p. Val6MetfsTer106 (MAF = 0.07%)[15]. The prevalence of MCD in Iceland is ~ 1/13,000[49]. We observed that 11 out of 16 homozygous carries of p.Ala128Val, and two out of three homozygous carriers of p.Val6MetfsTer106, have been diagnosed with hereditary corneal dystrophies (ICD10 code H18.5). We investigated the effect of these known disease variants on the corneal measures from the specular microscopy. P.Val6MetfsTer106 associates with cell density and HEX ($\beta = -3.02$ SD, $P = 4.5 \times 10^{-4}$ and $\beta = -2.00$ SD, $P = 9.3 \times 10^{-3}$, respectively) under the recessive model. The association is due to two homozygote carriers showing extremely low values (Supplementary Table 6.a). No homozygote carriers of p.Ala128val participated in the deCODE health study. We did not observe an effect of the two variants among heterozygous carriers (Supplementary Table 6b).

Out of the 6125 study participants, 194 (3.2%) had primary open-angle glaucoma (POAG). Among our corneal measures from the specular microscopy, POAG correlates most strongly with cell density (glaucoma patients have 104 cell/mm² lower cell density than controls, $P = 2.1 \times 10^{-6}$; F test). It also correlates with CH, IOPcc and CCT (Table 1).

Due to the correlation between various corneal measures and glaucoma status we assessed the effects of all 15 variants reported to associate with POAG[50] (Supplementary Data 2). First, we replicated the association of 11 out of the 15 variants with POAG ($P < 0.05$). The estimated replication power for the remaining four variants at *GMD2, ZFPM2, ATXN2,* and *PMM2* was > 99.7% at two-sided significance level of 0.05. We investigated the effects of the 11 replicated variants on different corneal measures (Fig. 4a). Five of the 11 variants associate with IOPg, where the POAG risk increasing allele associates with increased IOPg. Even though the strongest relationship of glaucoma is with CH (CH is lower in glaucoma patients), only two of the POAG variants associate with CH. Counter-intuitively, the *TMCO1* allele that increases glaucoma risk associates with lower CH but the *FNDC3B* allele that increases glaucoma risk associates with greater CH. The variant in *FOXC1* is the only POAG variant that also affects cell density. The correlation between the variants' effect on POAG and their effect on available corneal measures were not significant for any trait, controlling the false discovery rate at 0.05 (Supplementary Table 7).

**Previously reported variants for CCT and IOP**. GWASs have been published for two corneal traits, CCT and IOP. For CCT, 49 variants have been reported[7,11] and we replicate 28 of them ($P < 0.05$) (Supplementary Data 3). The CCT effects of these variants, and the two novel CCT variants in *ADAMTS17* and *GLTD82*, correlate with their effects on CH, CRF, and IOPg (Supplementary Table 7, Supplementary Figure 14). The effects on different corneal measures by the CCT increasing allele is shown in Fig. 4b. A recent study using participants from the UK Biobank ($N = 115,486$) identified 209 variants at 175 novel loci associating with IOPg[51]. We replicate 56 out of these 209 variants in the much smaller Icelandic data ($P < 0.05$) (Supplementary Data 4). The effects on different corneal measures by the IOPg increasing allele is shown in Fig. 4c. The effects of the IOP variants on IOPg correlate with their effects on CH, CRF, IOPcc, and POAG (Supplementary Table 7, Supplementary Figure 15).

## Discussion

Using measurements from a specular microscopy, we discovered seven novel variants associating with measurements of corneal structure, i.e., CCT, cell density, CV, and HEX. We further examined their effect on ocular biomechanics, such as CH, CRF, and IOP, as well as examining their effects on the risk of glaucoma and corneal diseases.

The most significant finding is the association of rs78658973 near *ANAPC1* with cell density. *ANAPC1* encodes the Anaphase Promoting Complex Subunit 1, a cell cycle-regulated E3 ubiquitin ligase that controls progression through mitosis and the G1 phase of the cell cycle[52]. The complex is composed of 15–17 subunits, which are highly conserved from yeast to humans. Mutations in the orthologous gene in the fruitfly, *shattered* (*shtd*), result in defective eye development[53] because of disruption of the G1 cell cycle arrest and progression through mitosis. Interestingly, rs78658973 also strongly increases CH, independently of its effect on cell density. CH measures ability of the corneal tissue to dampen pressure changes; such mechanical properties of most tissues are dominated by an extracellular matrix (ECM), in which the connective tissue fibers provide mechanical strength[54]. Many rare connective tissue disorders are characterized by both skeletal and eye abnormalities, such as Marfan syndrome, Ehlers–Danlos syndrome, dermatosparaxis type and Weill–Marchesani syndrome (OMIM: 154700, 225410, and 607511, respectively). An intron variant in *ANAPC1*, rs17040773[G], in complete LD with rs78658973[A] ($r^2 = 1.00$), has been reported to associate with decreased bone mineral density[55] and *ANAPC1* is expressed in bone (Hs.436527). A possible mechanism explaining the association of rs78658973 with cell density in the corneal endothelium may be the role *ANAPC1* has in controlling proliferation of the developing corneal endothelial cells, and likewise, proliferation of bone cells influencing bone density. However, a direct relationship between rs78658973 and *ANAPC1*, or any other gene at the locus, remains to be shown. The estimated fraction of variance of cell density explained by rs78658973 is 24% which is extremely high for such a complex human trait. In comparison, no other variant listed in the GWAS catalog[56] explains higher fraction of variance of a quantitative phenotype available in the extensive deCODE database (Supplementary Table 8). This finding has clinical importance, since reduction in endothelial cell density along with more variable cell size and shape is usually the first sign of corneal endothelial diseases and cell density is important when assessing risk of corneal endothelial failure prior to intraocular surgery[57]. Corneal graft survival studies found cell density lower than 1700 cells/mm², 6 months after graft surgery to be associated with increased risk of graft failure[58]. Thirty-year-old homozygote carriers of the *ANAPC1* variant have on average 2637 endothelial cells per mm², which is less than the average 70-year old noncarriers (Fig. 1e). Further studies could assess if carriers of the variant are more likely to suffer endothelial decompensation after intraocular surgery or if their cornea is associated with increased risk of corneal graft failure when used as donors.

We also found association of variants at known disease loci, *TCF4* and *CHST6* (FCED and MCD, respectively), with cell density and HEX, demonstrating how these structural measurements are affected by corneal diseases. The *TCF4* variant also strongly associates with CH and it is interesting to note that the *TCF4* and *ANAPC1* variants control almost the same quantitative corneal traits (Fig. 3) but have very different associations with corneal disease. Furthermore, glaucoma risk is strongly associated with lower cell density, while for example the *ANAPC1* variant that controls a quarter of the cell density variance does not associate directly with POAG or PACG. These findings suggest that the pathogenic processes that cause corneal diseases and

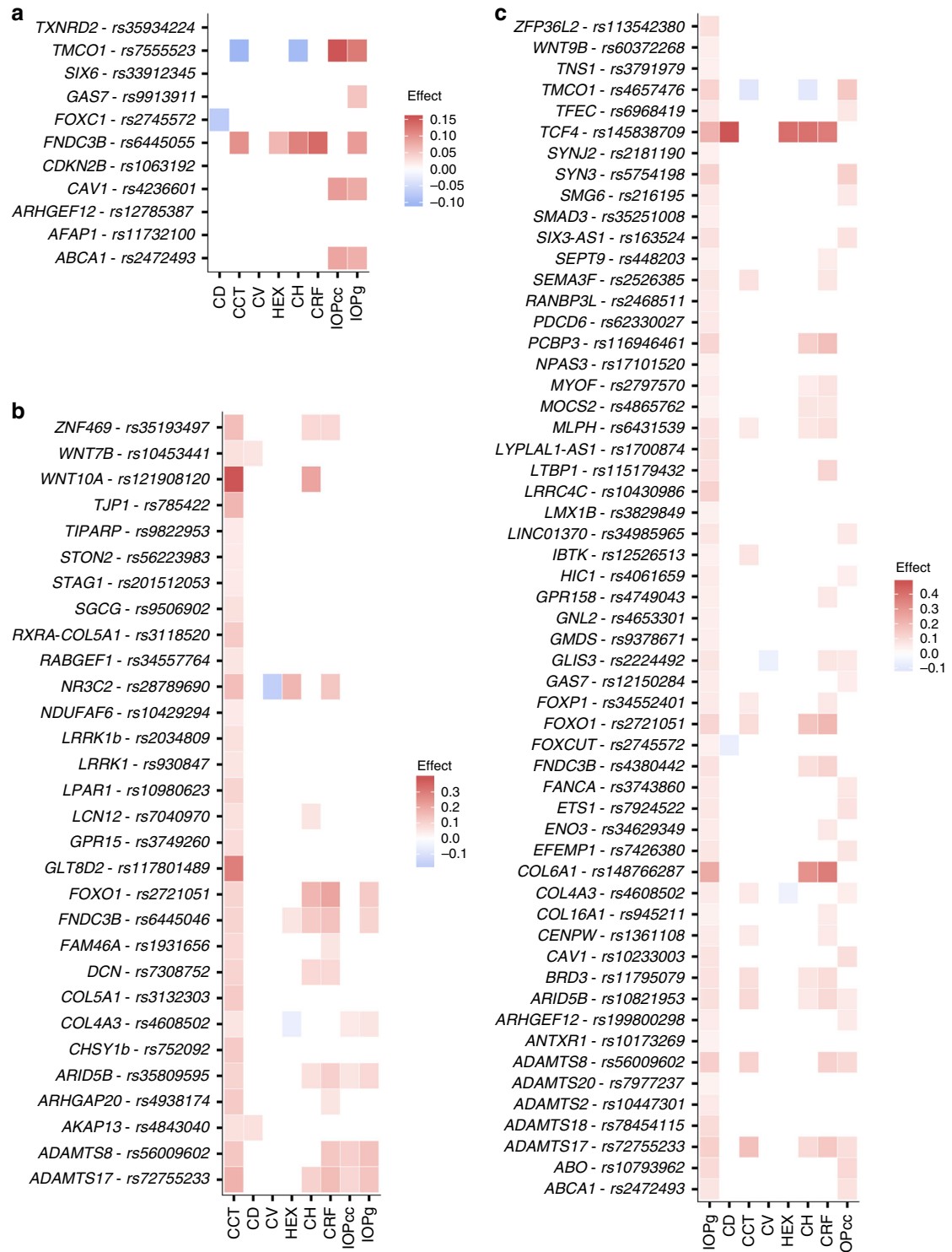

**Fig. 4** The effect of reported POAG, CCT, and IOP variants on corneal measures. Red color represents a positive effect on the corneal measures and blue color represents a negative effect. **a** Effect of previously reported POAG variants on corneal traits for the POAG risk increasing allele. Effects on the traits are shown for significant associations after adjusting for multiple testing with a false discovery rate procedure for each variant. **b** Effect of previously reported CCT variants that replicate in our data (P < 0.05) and novel CCT variants on corneal measures for the CCT increasing allele. Effects on other traits are shown for significant associations after adjusting for multiple testing with a false discovery rate procedure for each variant. **c** Effect of previously reported IOPg variants that replicate in our data (P < 0.05) on corneal measures for the IOPg increasing allele. Effects on other traits are shown for significant associations after adjusting for multiple testing with a false discovery rate procedure for each variant

glaucoma may also lower the cell density, but low cell density in and of itself does not increase risk of disease. In addition, we evaluated the effects of reported POAG variants on corneal structure to further explore the relationship between these traits. Even though CH, cell density, and CCT are lower in POAG patients, the effects of POAG associating variants on POAG risk do not correlate with their effects on these traits in our data. This suggests that these variants do not confer risk of POAG through their effect on these corneal metrics.

Healthy corneal endothelium is necessary for visual perception and its dysfunction is the most common reason for corneal transplantation. Our understanding of the structure and function of corneal endothelial cells and how they relate to diseases is limited. The work presented here constitutes a contribution toward shedding a light on this.

## Methods

**Study Subjects**. Endothelial images from non-contact auto-tracking and -focusing Konan CellCheck SL specular microscopy (Konan Medical USA Inc., Irvine, CA) and measures of ocular biomechanics using the Reichert ocular response analyzer[R] (ORA G3, Reichert Technologies, Depew, NY, USA) were obtained for 6266 Icelanders as a part of the deCODE health study. Participation in the deCODE health study also includes an online questionnaire and verbal interviews about health and lifestyle, a number of physical measurements, blood sample collection, and permission to access health-related information from a range of registries and records, including hospital data. We defined subjects in the study to have glaucoma if they reported history of glaucoma or had a hospital discharge diagnosis of primary open-angle glaucoma (ICD10 code H40.1). CCT associations were replicated in a previously described dataset from the Reykjavik Eye Study[59]. Testing variants for association with ocular diseases, we defined the glaucoma and corneal disorder populations based on six different ICD diagnoses; glaucoma information was obtained from participants in the Reykjavik Eye Study and from Icelandic ophthalmologists as described previously[60] (ICD10 H40.1, 4004 cases and 237,214 controls), primary open-angle glaucoma (ICD10 code H40.1, 1261 cases and 303,388 controls), primary angle-closure glaucoma (ICD10 code H40.2, 78 cases and 229,149 controls), disorders of cornea (ICD10 code H18, 133 cases and 255,274 controls), corneal degeneration (ICD10 code H18.4, 81 cases and 287,394 controls), and hereditary corneal dystrophies (ICD code H18.5, 119 cases and 301,665 controls). Written informed consent was obtained from all participants, in accordance with the Declaration of Helsinki, the study was approved by the Icelandic Data Protection Authority and the National Bioethics Committee (VSNb2015120006/03.01 with amendments).

The UK Biobank study is a large prospective cohort study of ~ 500,000 individuals from across UK, aged between 40 and 69 at recruitment[61]. Extensive phenotypic and genotypic information has been collected for the participants, including ICD coded diagnoses from inpatient and outpatient hospital episodes. In this study, we defined the glaucoma and corneal disorder populations based on six different ICD diagnoses; glaucoma (ICD10 code H40, 4428 cases and 404,139 controls), primary open-angle glaucoma (ICD10 code H40.1, 1035 cases and 402,449 controls), primary angle-closure glaucoma (ICD10 code H40.2, 699 cases and 407,868 controls), disorders of cornea (ICD10 code H18, 623 cases and 407,868 controls), corneal degeneration (ICD10 code H18.4, 118 cases and 396,627 controls), and hereditary corneal dystrophies (ICD code H18.5, 211 cases and 378,987 controls). Diagnoses were obtained from primary or secondary diagnoses codes a participant had recorded across all their episodes in hospital. Self-reported diagnoses were excluded from our analysis and we only included individuals determined to be of white British ancestry[62]. We did not exclude related individuals from the analysis but use LD score regression[63] to account for inflation in test statistics due to relatedness. In addition, height measurements were available for 407,825 individuals. UK Biobank's scientific protocol and operational procedures were reviewed and approved by the North West Research Ethics Committee (REC Reference Number: 06/MRE08/65), and informed consent was obtained from all participants.

**Images from Konan CellCheck SL specular microscopy**. The specular microscopy images were taken by specially trained nurses. One image was taken per eye. The nurse made a visual assessment of the image and selected automated analysis with auto trace for all normal images. Before automatic analysis, the size of cells (S, M, L, or XL) was determined and the cell border lines were compared with the cell borders visually.

All automated analysis were reviewed by a cornea specialist (G.M.Z.) and if found acceptable the automated analysis was used.

Images with any abnormalities, e.g., black areas, highly irregular cell size or shape, or poor quality images, were marked for manual analysis. These images were reviewed and analysed with the Center Method or in a few cases with the Flex Center Method. Images, where no cell structure was seen, were marked ungradeble. All analysis was done by a cornea specialist (G.M.Z.).

We excluded 415 low-quality images for 276 subjects prior to the analysis by regressing cell count against estimated cell density and removed images where the residuals from the fitted model where < −15. Thus, the sample size reduced from $N = 6266$ to $N = 6125$.

**Whole-genome sequencing**. The process used to whole-genome sequence the 28,075 Icelanders, and the subsequent imputation has been described in a recent publication[30,31]. In summary, we sequenced the whole genomes of 28,075 Icelanders using Illumina technology to a mean depth of at least 10 × (median 32 ×). SNPs and indels were identified and their genotypes called using joint calling with Graphtyper[64]. In total, 155,250 Icelanders were genotyped using Illumina SNP chips and their genotypes were phased using long-range phasing[65]. All sequenced individuals were also chip-typed and long-range phased, which provided information about haplotype sharing that was used to improve genotype calls. Genotypes of the 37.6 million high quality sequence variants were imputed into all chip-typed Icelanders. Using genealogic information, the sequence variants were also imputed into relatives of the chip-typed to further increase the sample size for association analysis and increased the power to detect associations. All of the variants that were tested had imputation information over 0.8.

In UK Biobank genotyping was performed using a custom-made Affimetrix chip, UK BiLEVE Axiom in the first 50,000 individuals[66], and with Affimetrix UK Biobank Axiom array in the remaining participants[67]; 95% of the signals overlap in both chips. Imputation was performed by Wellcome Trust Centre for Human Genetics using a combination of 1000Genomes phase 3[68], UK10K[69] and HRC reference panels[70], for up to 92,693,895 SNPs[62].

**Association analysis**. All quantitative ocular measurements were averaged for both eyes, rank-based inverse normal transformed to a standard normal distribution separately for each sex and adjusted for age using a generalized additive model[71]. For each sequence variant, a linear regression model, using the genotype as an additive covariate, the transformed quantitative trait as a response and assuming the variance–covariance matrix to be proportional to the kinship matrix, was used to test for association.

We used LD score regression to account for distribution inflation in the dataset due to cryptic relatedness and population stratification[63]. With a set of 1.1 million variants we regressed the $\chi^2$ statistics from our GWASs against LD score and used the intercepts as a correction factors. The estimated correction factors were 1.05, 1.03, 1.03, 1.06, 1.06, 1.05, 1.05, and 1.05 for cell density, CV, HEX, CCT, CH, CRF, IOPg, and IOPcc, respectively.

We used logistic regression to test for association between sequence variants and binary traits, regressing trait status against expected genotype count. In the Icelandic data, we adjusted for sex, age or age at death, county of birth, blood sample availability and an indicator function for the overlap of the subject's lifetime with the time span of phenotype collection, by including these variables in the logistic regression model. In the UK Biobank data, we adjusted for sex and age, as well as 40 principle components in order to adjust for population stratification.

For the meta-analysis we used a fixed-effects inverse variance method[72] based on effect estimates and standard errors from deCODE and the UK Biobank study. Sequence variants from deCODE and the UK Biobank imputation were matched on position and alleles.

**Significance thresholds**. The thresholds for genome-wide significance were estimated from the Icelandic data and corrected for multiple testing with a weighted Bonferroni adjustment using the enrichment of variant classes as weights with predicted functional impact among association signals[32]. With 37.6 million sequence variants in the Icelandic data, the weights given in Sveinbjornsson et al. were rescaled to control the family-wise error rate. This resulted in significance thresholds of $2.5 \times 10^{-7}$ for loss-of-function variants, $5.0 \times 10^{-8}$ for moderate-impact variants, $4.5 \times 10^{-9}$ for low-impact variants, $2.3 \times 10^{-9}$ for other variants within DHS sites, and $7.5 \times 10^{-10}$ for remaining variants. We evaluated false discovery rate, assessed with the $q$-value package in R. The $P$ value cutoff of $5.0 \times 10^{-8}$ corresponded to $q$-values of 0.0014 for cell density, 0.0035 for CV, 0.0117 for CCT, and 0.0116 for HEX, which add up to 3.4%.

When assessing, if associating variants have an effect on other corneal trait, we used the Benjamini–Hochberg false discovery rate (FDR) procedure controlling the FDR at 0.05 at each variant to account for multiple testing.

**Correlation between effect sizes**. We assessed the relationship between the effects of sequence variant on any two different traits by fitting a weighted linear regression model where the effects sizes for trait 1 was regressed on effect sizes for trait 2 and each variant was weighted by $f(1 − f)$ where $f$ is the minor allele frequency of the variants, so that rare variants have less weight in the computation than common variants. For binary traits we used log(OR) as effect size.

**Fraction of variance explained**. The fraction of variance explained is calculated using the formula $2f(1 − f)a^2$ where $f$ is the minor allele frequency and $a$ is the additive effect[73]. Calculating the fraction of variance explained for variants in the GWAS catalog, we estimated the effects of published variants with corresponding

phenotypes available in the deCODE data and calculated the fraction of variance explained using $f$ and $a$ obtained from the Icelandic population.

## Code availability

We used the following publicly available software for the whole-genome sequencing process:

for BWA 0.7.10 mem, see https://github.com/lh3/bwa; for Picard tools 1.117, see https://broadinstitute.github.io/picard/; for SAMtools 1.3, see http://samtools.github.io/; for Bedtools v2.25.0-76-g5e7c696z, see https://github.com/arq5x/bedtools2/; for GraphTyper 1.3, see https://github.com/DecodeGenetics/graphtyper; for Variant Effect Predictor, see https://github.com/Ensembl/ensembl-vep.

## Data availability

The sequence variants from the Icelandic population whole-genome sequence data have been deposited at the European Variant Archive under accession PRJEB15197, GWAS summary statistics for association with $P < 1 \times 10^{-6}$ are available in Supplementary Data 5. The authors declare that the data supporting the findings of this study are available within the article, it supplementary files, and upon request.

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

## Acknowledgements

We would like to thank the participants of the Icelandic deCODE health study for their valuable contribution to research. We also thank the staff at deCODE genetics recruitment center and core facilities, and all our colleagues, for their important collaboration on this work.

## Author contributions

E.V.I., S.B., G.Thorleifsson, P.S., A.O., U.S., G.M.Z., U.T., D.F.G., F.J., H.H. and K.S. designed the study and interpreted the results. E.V.I., S.B., S.K., G.Thorleifsson, G.A.A. and D.F.G. analyzed the data. E.V.I., H.H., G.Thorgeirsson, I.J., G.Thorleifsson, G.A.A., G.M.Z. and F.J. performed recruitment and phenotyping. The manuscript was drafted by E.V.I., S.B., P.S., D.F.G., F.J., H.H. and K.S. All authors contributed to the final version of the manuscript.

## Additional information

**Competing interests:** The authors E.V.I., S.B., G.Thorleifsson, P.S, A.O., U.S., S.K., G.A. A. G.Thorgeirsson, I.J., U.T., D.F.G., H.H., and K.S., declare competing financial interests as employees of deCODE genetics/Amgen, Inc. G.M.Z. and F.J. declare no competing interests.

