## [Peer Review File · Nature Communications]

Reviewer #1 (Remarks to the Author):

Key results

Ivarsdottir et al. performed genome-wide association analyses on the following four corneal structure measurements in 6,125 Icelanders: corneal endothelial cell density (CD), coefficient of cell size variation (CV), percentage of hexagonal cells (HEX) and central corneal thickness (CCT). Some of the explored measurements show high or moderate degrees of correlation. A total of ten independent risk variants were found to be associated with at least one of the four traits. Six of the 10 variants were stated to be novel in regard to the analyzed traits.

Follow up analyses included meta-analyses of ocular disease case control studies (e.g. glaucoma, corneal disorders) of the larger Iceland and the UK Biobank cohort. In addition, the authors explored the index variants and their effect on ocular biomechanics like corneal hysteresis, two intraocular pressure measures and the corneal resistance factor as well as investigated expression of implicated genes in ocular tissues of publicly available data.

Originality and significance

The presented analyses focused on several correlated ocular traits whose genetic factors were not extensively studied in genome-wide analyses before (except CCT). Six associations were stated to be novel for these traits, however the provided explorations revealed that none of these “novel” variants were directly relevant for any of the studied complex ocular disorders. Subjectively, the presented results might not be overly exciting for people who work in statistical, human or ocular genetics.

Data & methodology

The sampling mechanism for the 6,266 Icelanders of the discovery study is unclear. Overall, a well-established GWAS analysis strategy was applied. While correction factors for population stratification based on 1.1 million variants were provided for the four main traits, no QQ plots were included in the presented material. Providing QQ plots stratified by allele frequency would give peers a better sense about the applied adjustment for population stratification and about the behaviour of the test statistics across all frequency bins.

Generally, no analysis software packages were cited or stated in the manuscript thus making it hard to comprehend the performed analyses and its quality.

Appropriate use of statistics and treatment of uncertainties

Overall the authors provide confidence intervals and/or standard deviations for most of the reported analyses, but no confidence intervals are given for correlation analyses (e.g. Table 1).

The case control ratios of the analysed ocular traits (glaucoma traits and disorders of cornea; Supplementary Table 2) are extremely unbalanced and could potentially result in inflated test statistics if not addressed properly (Ma et al. Genet. Epidemiol 37, 539–550 (2013)). Without further

explanation or adjustment of these analyses, their appropriateness could be questioned because the case control ratios range from 59 to even >3500 controls per case.

The reported odds ratios of >140 for two rarer variants for hereditary corneal dystrophies in an autosomal recessive model (page 5) might be problematic and suffer from partial or complete separation (see Heinze and Schemper, Stat Med. 2002 Aug 30;21(16):2409-19).

The correction for multiple testing seems to be inconsistently applied throughout the manuscript: for discovery studies a correction for the number of independent variants was applied but not for the number of traits, for replication studies of known risk variants a correction for the number of traits but not for the number of variants was applied (bottom page 5, Figure 4, Supplementary Table 7,9,10).

Conclusions

The conclusion for the ANAPC1 variant that “no other variant reported in the GWAS catalog explains a higher fraction of variance of its associating quantitative phenotypes” needs to be taken with a grain of salt. First, the applied method warrants further explanation or references, secondly, the analysis was limited to measurements that were also available in deCODE data, and thirdly, the association might not be generalisable because only Icelanders were analysed.

Suggested improvements

The analyses on corneal hysteresis, two intraocular pressure measures and the corneal resistance factor seem to be doable genome-wide and consequently could be adjusted for population stratification. Presenting the corresponding Manhattan and QQ would help evaluate the quality of the obtained association results.

References

No references are given for some of the applied methods, e.g. correlation of between effect sizes or fraction of variance explained. The latter was probably described by Ju-Hyun Park et al. (Nature Genetics volume 42, pages 570–575 (2010); <https://www.nature.com/articles/ng.610>).

Clarity and context

The abstract and conclusions provided sufficient clarity and context. However, two paragraphs of the introduction and parts of the manuscript appear to focus on additional traits (MCD, FECD, IOP, CH, CRF). The main result section is interspersed with findings or literature from these additional phenotypes and appears disrupted in terms of flow, thereby affecting the clarity of the presented story.

Additional comments

1. Ordering Table 2 by trait instead of by p-value across all traits might improve its readability.
2. In general, no explanations are given for the bolded text in the tables (e.g. Table 1 and Supplement Tables 3 & 7-10).
3. The description of HEX and CCT and their corresponding Figure 1 in the second result paragraph seems incorrect/swapped.
4. The sample sizes are inconsistent throughout the manuscript: 6,123 (Table 1), 6,125 (results, Figure 2), 6,266 in methods. Understanding the applied exclusions would help understanding of these inconsistencies.
5. No imputation accuracy measures are given for the analysed variants; however, imputation information filter might have been applied (as indicated in the legend of Figure 2).
6. Figure 2 (especially 2A) would benefit from an axis break and labeled association peaks (locus name or rsID). Chromosome 23 should read X.
7. Figure 3,4 report gene names or chromosome bands while they are actually reporting single variants. Adding the variants reference IDs would be more appropriate and provide more clarity.
8. Effect sizes from linear and logistic regression analyses are mixed in Supplementary Table 3.
9. Gene name (CHST6) is missing for the two missense variants associated with MCD (center of page 5).
10. Gene names are not always given in italics (Table 2, Supplementary Tables)

Reviewer #3 (Remarks to the Author):

Ivarsdottir and co-authors present a genome-wide association study of 6,125 Icelanders on four corneal traits. These traits are:

- a) Corneal endothelial cell density
- b) Coefficient of cell size variation
- c) Percentage of hexagonal cells, and
- d) Central corneal thickness.

They identified multiple genetic loci showing strong evidence of association for the four corneal traits studied. Overall, it was a great pleasure reading the manuscript. Very interestingly, many of the loci showed pleiotropic effects across the four traits. There are both important positives which were emphasized, as well as important negatives (i.e. lack of association) which were also discussed

in the paper. The statistical analysis were beautifully controlled for, and appropriate adjustments using genomic control to account for dispersion of test statistics were undertaken.

I have some minor suggestions which could help in improving the presentation of the data to the general reader of Nature Communications.

1. The authors note clearly that glaucoma risk is strongly associated with decreased cell density ($P = 2.1 \times 10^{-6}$). However, whereas the index ANAPC1 SNP showed overwhelming association with cell density (as well as CV, HEX, and CH), it does not show any association at all with POAG and PACG. The authors should consider discussing this, as it could be very interesting to the general reader (a genetic variant which controls 24% of trait variance does not associate directly with the disease it is correlated with).

2. Strong evidence of association was observed between TCF4 and cell density as well as HEX, CH, and CRF. Very strikingly, TCF4 was also shown to be very strongly associated with hereditary corneal dystrophies, such as Fuch's. Apart from mentioning the association statistics, the authors should consider discussing this very interesting observation with a view of helping the general reader of the journal (as well as eye specialists) appreciate the results. In the opinion of the authors, why is the association between TCF4 and ANAPC1 so different with regards to disease? This is very interesting because they control almost the same corneal quantitative traits. In this regard, it is also curious that ZNF469, which the authors and others show to be very strongly associated with CCT and a gene for brittle cornea disease, showed almost no association at all with cell density.

3. Do any of the variants associate with Keratoconus?

Reviewer #1

Originality and significance

The presented analyses focused on several correlated ocular traits whose genetic factors were not extensively studied in genome-wide analyses before (except CCT). Six associations were stated to be novel for these traits, however the provided explorations revealed that none of these “novel” variants were directly relevant for any of the studied complex ocular disorders. Subjectively, the presented results might not be overly exciting for people who work in statistical, human or ocular genetics.

Response:

In our opinion, it is of great interest to reveal the causal relationship between correlated traits and diseases. Genetics is very useful in exploring the causal relations of phenotypes. One way of assessing the relationship between a biomarker/risk factor and disease, is by identifying a sequence variant that associates with the biomarker/risk factor of a disease of interest and testing if this variant associates directly with the disease. This method utilizes the fact that genotypes are randomly assigned at meiosis, and not affected by environmental risk factors or other related disease processes, and is thus independent of many confounders that are problematic in epidemiological studies. In this study we examine quantitative traits used clinically to diagnose corneal diseases and we see that endothelial diseases and glaucoma are associated with lower cell density. We find a sequence variant that associates highly significantly with cell density with large effect, accounting for a quarter of the population variance of cell density, but it has no association with disease. This effectively shows that low cell density in and of itself does not increase the risk of disease. However, further studies could assess if carriers of this variant are more likely to suffer endothelial decompensation after intraocular surgery or if their cornea is associated with increased risk of corneal graft failure when used as donors. We have now emphasized the clinical relevance of our findings in the discussion:

“This finding has clinical importance, since reduction in endothelial cell density along with more variable cell size and shape is usually the first sign of corneal endothelial diseases and cell density is important when assessing risk of corneal endothelial failure prior to intraocular surgery⁵⁷. Corneal graft survival studies found cell density lower than 1700 cells/mm² six months after graft surgery to be associated with increased risk of graft failure⁵⁸. Thirty year old homozygote carriers of the *ANAPCI* variant have on average 2.637 endothelial cells per mm² which is less than the average seventy year old non-carriers (Figure 1.e). Further studies could assess if carriers of the variant are more likely to suffer endothelial

decompensation after intraocular surgery or if their cornea is associated with increased risk of corneal graft failure when used as donors.

We also found association of variants at known disease loci, *TCF4* and *CHST6* (FCED and MCD, respectively), with cell density and HEX, demonstrating how these structural measurements are affected by corneal diseases. The *TCF4* variant also strongly associates with CH and it is interesting to note that the *TCF4* and *ANAPCI* variants control almost the same quantitative corneal traits (Figure 3) but have very different associations with corneal disease. Furthermore, glaucoma risk is strongly associated with lower cell density, while for example the *ANAPCI* variant that controls a quarter of the cell density variance does not associate directly with POAG or PACG. These findings suggest that disease processes that cause corneal diseases and glaucoma may also lower cell density, but low cell density in and of itself does not increase risk of disease.”

Data & methodology

The sampling mechanism for the 6,266 Icelanders of the discovery study is unclear.

Response:

We added to the main text (p. 4) that the deCODE health study is “a comprehensive phenotyping of a general population sample”.

Overall, a well-established GWAS analysis strategy was applied. While correction factors for population stratification based on 1.1 million variants were provided for the four main traits, no QQ plots were included in the presented material. Providing QQ plots stratified by allele frequency would give peers a better sense about the applied adjustment for population stratification and about the behaviour of the test statistics across all frequency bins.

Response:

We have added QQ-plots stratified by allele frequency as supplementary figures for all the traits (Supplementary Figures 8-11).

Generally, no analysis software packages were cited or stated in the manuscript thus making it hard to comprehend the performed analyses and its quality.

Response:

The software packages used for the association testing were developed in-house at deCODE and we describe our methods in detail. Regarding the methods used for sequencing the subjects and the subsequent imputation, we have now added more detailed information in methods:

“The process used to whole-genome sequence the 28,075 Icelanders, and the subsequent imputation has been described in a recent publication^{30,31}. We sequenced the whole genomes of 28,075 Icelanders using Illumina technology to a mean depth of at least 10X (median 32X). SNPs and indels were identified and their genotypes called using joint calling with the Genome Analysis Toolkit HaplotypeCaller (GATK version 3.4.07)⁶⁴. Genotype calls were improved by using information about haplotype sharing, taking advantage of the fact that all the sequenced individuals had also been chip-typed and long range phased. The 37.6 million variants passed the high quality threshold were then imputed into 155,250 Icelanders who had been genotyped with various Illumina SNP chips and their genotypes phased using long-range phasing⁶⁵. Using genealogic information, the sequence variants were imputed into relatives of the chip-typed to further increase the sample size for association analysis and increased the power to detect associations. All of the variants that were tested had imputation information over 0.8.”

Appropriate use of statistics and treatment of uncertainties

Overall the authors provide confidence intervals and/or standard deviations for most of the reported analyses, but no confidence intervals are given for correlation analyses (e.g. Table 1).

Response:

We have added a supplementary table 1 with p-values for the correlation coefficients shown in Table 1.

The case control ratios of the analysed ocular traits (glaucoma traits and disorders of cornea; Supplementary Table 2) are extremely unbalanced and could potentially result in inflated test statistics if not addressed properly (Ma et al. Genet. Epidemiol 37, 539–550 (2013)). Without further explanation or

adjustment of these analyses, their appropriateness could be questioned because the case control ratios range from 59 to even >3500 controls per case.

Response:

As we describe in methods, in our case-control association testing we use likelihood ratio tests. The paper the reviewer refers to states that for high count variants the likelihood ratio test is well-calibrated, even in unbalanced studies. However, they also state that for low-count variants, the likelihood ratio test can be slightly anti-conservative for some MACs (minor allele counts). We find this statement to be in paradox with Figures 1.C and 2.A and B in the paper, which actually show that the likelihood ratio test is performing well for unbalanced studies (see snap-shot from paper below). They also state that all test perform well for variants with $MAC > 400$ and suggest using that as a threshold. The variants that we are testing in our study all have high MAC. The least frequent variant (at *GLT8D2*) has $MAC > 13000$. We therefore are not worried that our performed tests are anti-conservative. Also, all of the case-control analysis were performed on the genome-wide level and we used LD score regression to account for inflation of test statistics.

Figure 1:

Figure 2:

The reported odds ratios of >140 for two rarer variants for hereditary corneal dystrophies in an autosomal recessive model (page 5) might be problematic and suffer from partial or complete separation (see Heinze and Schemper, *Stat Med.* 2002 Aug 30;21(16):2409-19).

Response:

We agree that odds ratios for such rare variants causing a disease are not meaningful. We have thus decided to report instead the number of homozygotes for each variant that has been diagnosed with the disease:

“Two different mutations in *CHST6* are known to cause MCD in Iceland; a missense variant p.Ala128Val (MAF=0.66%) and a frameshift variant p.Val6MetfsTer106 (MAF=0.07%)¹⁶. The prevalence of MCD in Iceland is $\sim 1/13,000$ ⁴⁹. We observed that 11 out of 16 homozygous carriers of p.Ala128Val, and two out of three homozygous carriers of p.Val6MetfsTer106, have been diagnosed with hereditary corneal dystrophies (ICD10 code H18.5).”

The correction for multiple testing seems to be inconsistently applied throughout the manuscript: for discovery studies a correction for the number of independent variants was applied but not for the number

of traits, for replication studies of known risk variants a correction for the number of traits but not for the number of variants was applied (bottom page 5, Figure 4, Supplementary Table 7,9,10).

Response:

In the discovery studies, we think it is not appropriate to correct for the number of traits studies. The number of significant findings in each phenotype is substantial which translates into a false discovery rate of 0.0014 for cell density, 0.0035 for CV, 0.0117 for CCT and 0.0116 for HEX at the 5.0×10^{-8} significance threshold. These false discovery rates, which add up to 3.4%, suggest that our multiple testing procedure is overall conservative. FDR approach using 5% discovery rate has been suggested robust in a previous genome-wide association publication (Nelson, C.P., et.al., Nature Genetics volume 49, pages 1385–1391 (2017)). We have added a statement describing this to the text of the manuscript: “We evaluated false discovery rate, assessed with the qvalue package in R. The P value cutoff of 5.0×10^{-8} corresponded to q-values of 0.0014 for cell density, 0.0035 for CV, 0.0117 for CCT and 0.0116 for HEX, which add up to 3.4%.”

When assessing if any of the 10 GWS variants associate with an ocular disease we adjust for the number of tests performed by adjusting for the number of phenotypes and number of variants.

In the heatmap–figures, where we are exploring how the detected variants associate with different corneal measures we think it is not necessary to adjust for the number of variants. We are more exploring association patterns than claiming novel associations (unless they are genome-wide significant). But instead of using Bonferroni correction procedure we have now switched to using the Benjamini-Hochberg false discovery rate (FDR) procedure controlling the FDR at 0.05 at each variant to account for multiple testing. We added to methods:

“When assessing if associating variants have an effect on other corneal traits, we used the Benjamini-Hochberg false discovery rate (FDR) procedure controlling the FDR at 0.05 at each variant to account for multiple testing.”

Conclusions

The conclusion for the ANAPC1 variant that “no other variant reported in the GWAS catalog explains a higher fraction of variance of its associating quantitative phenotypes” needs to be taken with a grain of salt. First, the applied method warrants further explanation or references, secondly, the analysis was limited to measurements that were also available in deCODE data, and thirdly, the association might not be generalisable because only Icelanders were analysed.

Response:

We have changed the sentence to “In comparison, no other variant listed in the GWAS catalog⁵⁶ explains higher fraction of variance of a quantitative phenotype available in the extensive deCODE database“ and we have added reference in the methods for the formula used to compute the fraction of variance explained.

We can't replicate the association between the *ANAPCI* variant and cell density abroad due to lack of data. To the best of our knowledge, quantitative measures from a specular microscopy have never been collected before for a genetic study. However, Icelanders have proven a reasonably good animal model for *Homo sapiens*. Also, the UK Biobank has CH measures and the association we see between *ANAPCI* and CH in our data (rs78658973; $\beta=0.19$, $P=2.6 \times 10^{-19}$) replicates well in the UK Biobank data (rs1044864; $\beta=0.12$, $P=2.6 \times 10^{-100}$, $R^2=1.0$ between rs1044864 and rs78658973, <https://biobankengine.stanford.edu/variant/2-112526866>). We therefore believe that the cell density association is also not restricted to Icelanders.

Suggested improvements

The analyses on corneal hysteresis, two intraocular pressure measures and the corneal resistance factor seem to be doable genome-wide and consequently could be adjusted for population stratification. Presenting the corresponding Manhattan and QQ would help evaluate the quality of the obtained association results.

Response:

The analysis of CH, CRF, IOPg and IOPcc were performed genome-wide and reported p-values have been adjusted for population stratification. We have now added QQ and Manhattan plots in the supplement (Supplementary Figures 5-6) as well as this information in the methods: “The estimated correction factors were 1.05, 1.03, 1.03, 1.06, 1.06, 1.05, 1.05 and 1.05 for cell density, CV, HEX, CCT, CH, CRF, IOPg and IOPcc, respectively.”

References

No references are given for some of the applied methods, e.g. correlation of between effect sizes or fraction of variance explained. The latter was probably described by Ju-Hyun Park et al. (Nature Genetics volume 42, pages 570–575 (2010); <https://www.nature.com/articles/ng.610>).

Response:

We have added a reference for the fraction of variance explained method (Gudbjartsson, D. F. et al. Many sequence variants affecting diversity of adult human height. Nat. Genet. 40, 609–615 (2008)).

A simple weighted linear regression (Mandel, J., The Statistical Analysis of Experimental Data. (1964)) is used to calculate the correlation between effect sizes which is a general method that we describe in the methods chapter. For clarity we added a more detailed explanation for the chosen weights:

“.., so that rare variants have less weight in the computation than common variants”

Clarity and context

The abstract and conclusions provided sufficient clarity and context. However, two paragraphs of the introduction and parts of the manuscript appear to focus on additional traits (MCD, FECD, IOP, CH, CRF). The main result section is interspersed with findings or literature from these additional phenotypes and appears disrupted in terms of flow, thereby affecting the clarity of the presented story.

Response:

We have added and rephrased sentences in the introduction in the attempt to make the flow better and we have added subtitles in the result section to make the presented story clearer.

“The measures of endothelial structure are clinically used to diagnose corneal diseases. Several corneal diseases including...

Cell density in the corneal endothelium may be reduced in glaucoma patients²¹. Glaucoma is an ocular disease...”

Additional comments

1. Ordering Table 2 by trait instead of by p-value across all traits might improve its readability.

Response:

We agree and table 2 is now ordered by trait first and then p-value.

2. In general, no explanations are given for the bolded text in the tables (e.g. Table 1 and Supplement Tables 3 & 7-10).

Response:

Explanations for bolded text has now been added in table legends.

3. The description of HEX and CCT and their corresponding Figure 1 in the second result paragraph seems incorrect/swapped.

Response:

This has now been fixed.

4. The samples sizes are inconsistent throughout the manuscript: 6,123 (Table 1), 6,125 (results, Figure 2), 6,266 in methods. Understanding the applied exclusions would help understanding of these inconsistencies.

Response:

There was a typing error in table 1, the sample size is 6,125. In methods we describe how we exclude low-quality images for 415 eyes in 276 individuals. For clarity we added in methods on page 15: “Thus, the sample size reduced from N=6,266 to N=6,125.”

5. No imputation accuracy measures are given for the analysed variants; however, imputation information filter might have been applied (as indicated in the legend of Figure 2).

Response: We have added a sentence in the legend of table 2:

“The imputation information for all the detected variants is >0.99.”

6. *Figure 2 (especially 2A) would benefit from an axis break and labeled association peaks (locus name or rsID). Chromosome 23 should read X.*

Response: The manhattan figures now have labelled peaks by locus name and figure 2A has axis break.

7. *Figure 3,4 report gene names or chromosome bands while they are actually reporting single variants. Adding the variants reference IDs would be more appropriate and provide more clarity.*

Response:

We have now added rs-names in the figures.

8. *Effect sizes from linear and logistic regression analyses are mixed in Supplementary Table 3.*

Response:

This has been fixed.

9. *Gene name (CHST6) is missing for the two missense variants associated with MCD (center of page 5).*

Response:

This has been fixed.

10. *Gene names are not always given in italics (Table 2, Supplementary Tables)*

Response:

This has been fixed.

Reviewer #3 (Remarks to the Author):

1. The authors note clearly that glaucoma risk is strongly associated with decreased cell density ($P = 2.1 \times 10^{-6}$). However, whereas the index ANAPC1 SNP showed overwhelming association with cell density (as well as CV, HEX, and CH), it does not show any association at all with POAG and PACG. The authors should consider discussing this, as it could be very interesting to the general reader (a genetic variant which controls 24% of trait variance does not associate directly with the disease it is correlated with).

2. Strong evidence of association was observed between TCF4 and cell density as well as HEX, CH, and CRF. Very strikingly, TCF4 was also shown to be very strongly associated with hereditary corneal dystrophies, such as Fuch's. Apart from mentioning the association statistics, the authors should consider discussing this very interesting observation with a view of helping the general reader of the journal (as well as eye specialists) appreciate the results. In the opinion of the authors, why is the association between TCF4 and ANAPC1 so different with regards to disease? This is very interesting because they control almost the same corneal quantitative traits. In this regard, it is also curious that ZNF469, which the authors and others show to be very strongly associated with CCT and a gene for brittle cornea disease, showed almost no association at all with cell density.

Response to 1 and 2:

We have emphasized the points made by the reviewer in the discussion:

“The *TCF4* variant also strongly associates with CH and it is interesting to note that the *TCF4* and *ANAPC1* variants control almost the same quantitative corneal traits (Figure 3) but have very different associations with corneal disease. Furthermore, glaucoma risk is strongly associated with lower cell density, while for example the *ANAPC1* variant that controls a quarter of the cell density variance does not associate directly with POAG or PACG. These findings suggest that disease processes that cause corneal diseases and glaucoma may also lower cell density, but low cell density in and of itself does not increase risk of disease.”

3. Do any of the variants associate with Keratoconus?

Response:

We did not include keratoconus before in our analysis due small number of cases (127 cases combined in Iceland and UK Biobank). But we have now performed the meta-analysis and found that none of the variants associate with keratoconus. We have added these results to Supplementary Tables 3 and 4.

Reviewer #3 (Remarks to the Author):

The authors have been very responsive to the previous round of review. The statistical analysis of the genetic data is transparently presented, both in the main text as well as in the Supplementary Tables and Figures.

The addition of data on keratoconus patients and controls is helpful despite the small number (N=127 cases). This is because as the authors now show in Supplementary Table 4, they again observe the paradoxical effect first reported in 2013 (Lu Y et al., Nat Genet 45:155-163) whereby the largest effect CCT locus at ZNF469 which decreases CCT confers decreased susceptibility to keratoconus. One would expect a gene locus which **decreases** CCT to **increase** risk of keratoconus, so this was unexpected. The authors here report the same trend (OR = 0.73, P = 0.027). This kind of supportive evidence provided in the context of previously 'controversial' findings helps clarify the literature and guides future work in this field.

I have no further concerns.

Reviewer #4 (Remarks to the Author):

I think the authors have addressed the reviewers comments adequately.